# Retrofitting Light-weight Language Models for Emotions using Supervised Contrastive Learning

**Sapan Shah♣♠, Sreedhar Reddy♠, and Pushpak Bhattacharyya♣**
♠TCS Research, Tata Consultancy Services, Pune
♣Indian Institute of Technology Bombay, Mumbai
{sapan.hs,sreedhar.reddy@tcs.com}
pb@cse.iitb.ac.in

## Abstract

We present a novel retrofitting method to induce emotion aspects into pre-trained language models (PLMs) such as BERT and RoBERTa. Our method updates pre-trained network weights using contrastive learning so that the text fragments exhibiting similar emotions are encoded nearby in the representation space, and the fragments with different emotion content are pushed apart. While doing so, it also ensures that the linguistic knowledge already present in PLMs is not inadvertently perturbed. The language models retrofitted by our method, i.e., BERTEmo and RoBERTaEmo, produce emotion-aware text representations, as evaluated through different clustering and retrieval metrics. For the downstream tasks on sentiment analysis and sarcasm detection, they perform better than their pre-trained counterparts (about 1% improvement in F1-score) and other existing approaches. Additionally, a more significant boost in performance is observed for the retrofitted models over pre-trained ones in few-shot learning setting.

## 1 Introduction

Despite the emergence of powerful models like GPT4 and LaMDA, lightweight pre-trained language models (PLMs) such as BERT and RoBERTa remain relevant since they are computationally less expensive and open. Being trained on massive amount of data available over the web, these models are extremely good at encoding general language properties, resulting in highly accurate word, sentence, and paragraph representations. These representations are then typically fine-tuned for given tasks using task-specific datasets. However, the representations learned using these PLMs are general purpose, and they do not capture the affective aspects (such as emotions, affects, etc.) of human communication well. Consider a few exemplar pairs of text fragments, along with the emotion they evoke, in Table 1. We expect the pairs of frag-

ments with the same emotion label to have similar embeddings if the PLMs are cognizant of emotion content. However, as evident from Table 1, the cosine distance (computed using BERT [CLS] embeddings) between pairs exhibiting the same emotion is anomalously higher than those between different emotion categories. This observation suggests that PLMs do not adequately capture emotion aspects that are inherently present in natural language text. Incorporating them into PLM representations can significantly benefit NLP applications that are affective in nature, such as sentiment analysis, sarcasm detection, empathetic agents, etc.

One straightforward way to inject emotions into PLMs is through transfer learning by fine-tuning them on emotion recognition task. Multi-task learning is also a potential alternative wherein the affective task of interest is paired with the emotion recognition task and optimized jointly. However, these approaches are not that effective in capturing emotion aspects (see section 5). The community has focussed on using resources such as sentiment lexicons, emoticons, etc., to impart affective aspects into PLMs, by modifying pre-training objectives such as masked language modeling (Zhou et al., 2020; Aduragba et al., 2021) and next sentence prediction (Babanejad et al., 2020). These approaches either learn PLMs from scratch or continually pre-train them on domain-specific corpora. A few works (Suresh and Ong, 2021; Yin et al., 2020) have also explored attention-based network modules over contextualized embeddings to capture affective semantics. The approaches mentioned above, however, are highly sensitive to the training corpus and the lexical resources used, and do not generalize well across tasks. Retrofitting methods that use external knowledge to improve representations have been well explored in the community for static embeddings (Mrkšić et al., 2016; Shah et al., 2020). There are also attempts to retrofit PLMs to learn robust contextualized word embeddings

| Text fragment 1 | Text fragment 2 | BERT | Ours |
|---|---|---|---|
| Worst advice ever. This would piss me off. *-anger* | She sounds abusive and narcissistic. Delete your facebook, hit the gym and kill her. *-anger* | 0.2013 | 0.1085 |
| | I love how this sub is gender less *-joy* | 0.0702 | 0.5254 |
| I'm so glad I watched this entire thing. That's incredible. *-joy* | Glad to see we are still around :). *-joy* | 0.1063 | 0.1746 |
| | I'm sorry it happened like that for you. I really can't imagine. *-sadness* | 0.082 | 0.3429 |

Table 1: Cosine distance between pairs of text fragments exhibiting same and different emotions. With BERT, the distances between text fragments are not in agreement with their emotion content. Our method fixes this by retrofitting BERT for emotions (Ours = BERTEmo)

(Shi et al., 2019), sentence embeddings (Cai et al., 2022), etc. However, PLM retrofitting is relatively less explored for human affects domains.

Recently, contrastive learning has been actively explored in self-supervised setting to learn better sentence representations using data augmentation (Gao et al., 2021; Fang et al., 2020). Supervised contrastive learning (SCL) (Khosla et al., 2020) improves it further by generating informed training data using label information. SCL has been shown to learn robust sentence classification models (Gunel et al., 2021; Sedghamiz et al., 2021). It has also been applied to affective tasks such as arousal classification (Pinitas et al., 2022), emotion recognition in conversation (Song et al., 2022), and so on. A common theme across all these methods is that SCL has been applied only in a single task setting. Despite being fundamentally a representation learning tool, it has neither been explored in transfer learning nor in retrofitting settings.

In this work, we present a novel retrofitting method to learn emotion-aware PLMs using supervised contrastive learning as a transfer learning tool. We use go_emotions (Demszky et al., 2020), the largest publicly available emotion recognition dataset, as a retrofitting corpus. Our method updates PLM network weights such that the text fragments exhibiting similar emotions are encoded nearby in the representation space, and fragments with different emotion content are pushed apart. While doing so, it also ensures that the linguistic knowledge originally present in the PLM is preserved. We refer to the resulting emotion-aware models as ∗Emo[1]. Our contributions are:

1. A novel retrofitting method to learn emotion-aware PLMs using supervised contrastive learning (section 4). The ∗Emo models produce emotion-aware sentence representations

---

[1] We use ∗Emo to refer to both BERTEmo and RoBERTaEmo combinedly.

ensuring that sentences with similar emotions have high cosine similarity and sentences with different emotions have low similarity. The sentence embeddings from BERT and RoBERTa do not have this property (quantitatively shown in Table 3).

2. A detailed evaluation (Table 6) showing that the ∗Emo models perform better than their pre-trained counterparts (about 1% statistically significant improvement in F1-score) and other approaches, such as transfer learning and multi-task learning, on sentiment analysis and sarcasm detection tasks. In limited data setting, they perform exceedingly better than BERT and RoBERTa, as exemplified by the few-shot learning experiments on sentiment analysis task (Figure 2).

## 2 Related Work

### 2.1 Affect-aware PLMs

Resources such as sentiment lexicons, emoticons, etc., have been actively used to learn affect-aware PLMs by updating masked language modeling (MLM) objective. For instance, SentiX (Zhou et al., 2020) learns the BERT model from scratch using reviews/ratings (Yelp, Amazon) dataset by increasing the masking probability for sentiment words and emoticons; EmoBERT (Aduragba et al., 2021) continually pre-trains by masking emotion-bearing words in tweet dataset; and so on. Unlike MLM, Babanejad et al. (2020) generate emotion-feature vector using EmoLex (Mohammad and Turney, 2013) and use it for the next sentence prediction (NSP) task in continual pre-training setting. CARER (Saravia et al., 2018) is a graph-based approach for learning emotion-aware contextualized representations. SentiBERT (Yin et al., 2020) adds an attention network based semantic composition module over BERT representations, and fine-tunes

it on sentiment analysis task using constituency trees. A few approaches use affective resources directly during end-task fine-tuning. For instance, KEA (Suresh and Ong, 2021) uses token-level valence, arousal, and dominance scores to learn attention weighted sentence representations. The affect-aware approaches described above, however, are highly sensitive to the training corpus and the lexical resources used, and may not generalize well across tasks.

Retrofitting is a post-processing method that updates pre-trained network weights to respect constraints extracted from external knowledge resources. It is well explored for static embeddings, e.g., retrofitting for relations such as synonymy, and hypernymy (Mrkšić et al., 2016; Shah et al., 2020), for affective lexicons (Shah et al., 2022b,a), and so on. Shi et al. (2019) retrofit PLM in ELMo using paraphrase context to learn robust contextualized word embeddings. Other notable approaches include retrofitting sentence embeddings with abstract meaning representations in multilingual setting (Cai et al., 2022), learning label-aware conditional mask language model for contextual data augmentation (Wu et al., 2018), and so on. PLM retrofitting, however, is relatively less explored for the human affects domain.

## 2.2 Large Language Models

The research community has recently been actively applying large language models (LLMs), especially ChatGPT, to sentiment analysis tasks (Zhong et al., 2023; Wang et al., 2023). These works have explored a variety of approaches, including zero-shot and in-context learning, as well as prompting methods such as chain-of-thought (CoT). Although these works have considered aspects such as polarity shift detection, aspect-based analysis, and sentiment inference, for the standard sentiment classification task, they experiment only with the Stanford sentiment treebank binary classification (SST2 dataset). While the reported results are comparable to fine-tuned BERT and RoBERTa, the SST2 dataset is a more straightforward, coarse-grained classification task with a SOTA accuracy of 97.5%. A more detailed evaluation with fine-grained sentiment classes, non-standard English, etc., is required before establishing the efficacy of LLMs. To this end, Zhang et al. (2023) perform an exhaustive set of experiments with ChatGPT using zero-shot and in-context learning on 13 sentiment

analysis tasks. Unlike using LLMs during inference, Deng et al. (2023) use ChatGPT with CoT prompt to obtain labeled data for sentiment analysis and then use the weakly labeled data to learn an accurate student model. This approach, however, has only been tested on social media content in the finance domain.

## 2.3 Contrastive Learning

Contrastive learning (CL) aims to learn an embedding space such that similar data points are mapped close to each other while dissimilar points are pushed apart. Self-supervised contrastive learning uses data augmentation techniques such as lexical editing (Wu et al., 2018), back translation (Fei et al., 2020; Fang et al., 2020), dropout (Gao et al., 2021), cut-off (Shen et al., 2020), etc., to generate similar (positive) data points for the given anchor. The dissimilar (negative) points for the anchor are then selected randomly from in-batch examples. Supervised contrastive learning (SCL) (Khosla et al., 2020) takes this idea further by using labeled data to generate positives from the same class as the anchor, thereby providing more variability than data augmentation. SCL has been recently applied to text classification problems, e.g., joint learning with SCL and cross-entropy (CE) loss (Gunel et al., 2021), SCL followed by standard fine-tuning using CE in a pipeline fashion (Sedghamiz et al., 2021). It has also been applied for emotion recognition in conversations (Song et al., 2022), affect-infused representations for arousal classification (Pinitas et al., 2022), and so on. However, the approaches described above have applied SCL only in a single task setting. In contrast, we explore SCL as a transfer learning tool to induce emotion aspects into PLMs in retrofitting setting.

## 3 Retrofitting Corpus: go_emotions

To retrofit PLMs for emotions, we need a corpus that provides emotion annotations for text fragments, i.e., emotion recognition datasets. Such datasets, varying in size, labeling scheme, domain, etc., have been proposed in the field (see Bostan and Klinger 2018 for review). Being the largest publicly available dataset, we use go_emotions (Demszky et al., 2020) in this work. It provides fine-grained emotion annotations (27 categories) for 54,263 English Reddit comments. The dataset has been carefully created to avoid offensive, identity and religion terms. Though the text fragments

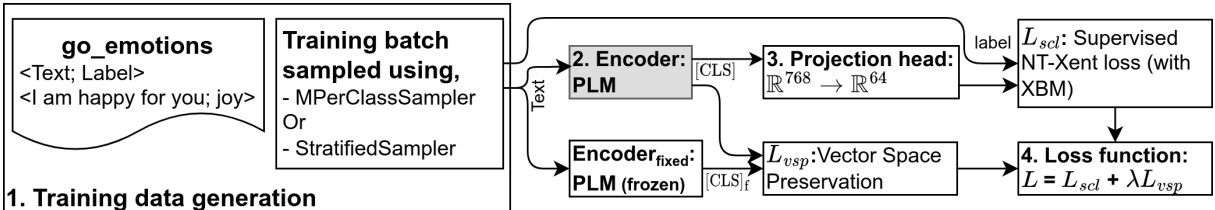

Figure 1: Architecture for learning emotion-aware PLMs in retrofitting setting

| Number of examples | 58,009 |
|---|---|
| Number of raters per example | 3 |
| Number of examples with 2+ raters agreeing on at least 1 label | 54,263 |
| Number of emotions | 27+Neutral |
| Number of examples with unique labels | 45,446 |

Table 2: go_emotions dataset: examples with single emotion label are used as a retrofitting corpus

in the dataset are marked with multiple emotion categories, we only use a subset of 45,446 examples that are annotated with a single emotion label. Table 2 shows the summary statistics for go_emotions.

## 4 Retrofitting Method

The application of supervised contrastive learning in our retrofitting method is inspired by (Khosla et al., 2020; Sedghamiz et al., 2021), albeit applied in a transfer learning setting. Our method also ensures that the linguistic knowledge already present in PLM network weights is not inadvertently perturbed. Figure 1 shows the architecture for our retrofitting method.

**1. Training data generation:** We sample training examples from go_emotions to create a minibatch B of size N (i.e., $\{x_i, y_i\}_{i=1..N}$; $x_i$ = text fragment; $y_i$ = label). We consider two sampling alternatives: (1) **MPerClassSampler** samples an equal number of examples from each emotion category; (2) **StratifiedSampler** samples examples from an emotion category proportional to the total number of examples of that category in go_emotions.

**2. Encoder:** The PLM under consideration is treated as an encoder function. It takes text fragments $x_i$ from batch B as input and returns the $[\text{CLS}]_i$ embeddings as output, i.e., $\text{Enc}(x_i) = [\text{CLS}]_i$.

**3. Projection head:** It maps $[\text{CLS}]_i$ to

an $l_2$-normalized vector $z_i \in \mathbb{R}^d$, i.e., $\text{Proj}([\text{CLS}]_i) = z_i$. We consider two variants for Proj: (1) Linear: $[\text{CLS}]_i$ projected linearly to $\mathbb{R}^{64}$; (2) MLP: $[\text{CLS}]_i$ projected non-linearly to $\mathbb{R}^{64}$ using a single hidden layer of size 768.

**4. Loss function:** In self-supervised contrastive learning, the normalized temperature-scaled cross entropy (NT-Xent) loss proposed by Chen et al. (2020) has been shown to learn robust latent representations by implicitly providing hard positive/negative mining. Khosla et al. (2020) extended it further for SCL by incorporating label information into the loss function. We use the supervised version of the loss function[2] which is as follows,

$$L_{scl} = \sum_{i=1}^{N} \frac{-1}{|P(i)|} \sum_{p \in P(i)} \log \frac{e^{(z_i \cdot z_p)/\tau}}{\sum_{b \in B(i)} e^{(z_i \cdot z_b)/\tau}} \quad (1)$$

Here, B(i): $b \in B \setminus \{i\}$; P(i): $\{p \in B; y_p = y_i\}$; and $\tau$: temperature scaling factor. Setting $\tau$ to a low value gives more importance to hard positives/negatives, whereas a high value weighs all pairs nearly equally.

*Vector space preservation (VSP):* Being trained on a vast amount of data, PLMs are extremely good at encoding lexical, syntactic, and semantic relations, concept similarities, and so on (Jawahar et al., 2019; Lin et al., 2019). We do not want to lose such valuable information while retrofitting them for emotions. While retrofitting literature on static embeddings has recognized and addressed this issue by introducing a regularization term that preserves the topology of pre-trained vector space (Mrkšić et al., 2016; Mrkšić et al., 2017), such regularization has not been considered by retrofitting methods for PLMs. To address this, we use the following regularization term,

$$L_{vsp} = \sum_{i=1}^{N} \|\text{Enc}(x_i) - \text{Enc}_{\text{fixed}}(x_i)\|_2 \quad (2)$$

---

[2]Contrastive learning on deep networks generally requires large mini-batch sizes for training stability (Radford et al., 2021; Chen et al., 2020). To support this, we use cross-batch memory (XBM) (Wang et al., 2020) with the SCL loss.

While $\text{Enc}(\cdot)$ comes from step 2, $\text{Enc}_{\text{fixed}}(\cdot)$ is a fixed version where the network weights from PLM are frozen, providing the snapshot of the PLM before the application of contrastive learning. With this regularization, the network weights in $\text{Enc}(\cdot)$ are updated such that the sentence embeddings in [CLS] do not deviate much from their pre-trained version. The final loss function used by our method is then: $L = L_{scl} + \lambda L_{vsp}$, where $\lambda$ is a hyper-parameter that determines how strictly the original pre-trained vector space is preserved.

Post training, we discard the projection head. While the encoder $\text{Enc}(\cdot)$, with its updated weights, provides us with emotion-aware PLM.

## 5 Experiments

We evaluate our retrofitting method on two PLMs in BERT (Devlin et al., 2019) and RoBERTa (Liu et al., 2019) {base, uncased versions}. The emotion-aware PLMs retrofitted by our method are referred to as ∗Emo, i.e., BERTEmo and RoBERTaEmo. To select the best hyper-parameter configuration, we consider two aspects: (1) quality of ∗Emo in learning emotion-aware embeddings, measured using clustering and retrieval metrics (refer section 5.2 for details) (2) vector space preservation: mean cosine distance between sentence embeddings obtained from ∗Emo and its pre-trained version (cosine distance = 1 - cosine similarity; Range=[0, 2]). We first filter vector space preserving configurations. From the filtered set, we then choose the configuration with the highest AMI (clustering quality metric) as the best configuration. We detail this process and report the complete hyper-parameter grid search in Appendix A.

### 5.1 Compared Work

The BERT and RoBERTa models are considered as **pre-trained** baseline. For downstream tasks, we add a linear classification head over the [CLS] embeddings and jointly fine-tune both the PLM network weights and the classification head using cross-entropy (CE) loss (referred to as standard fine-tuning).

For transfer learning (**TLearn**), we first add a classification head over the [CLS] embeddings and perform fine-tuning on the emotion recognition task using the go_emotions dataset. After training for emotions, the updated PLM is further fine-tuned on end tasks using a linear classification head. For multi-task learning (**MTL**), we add two classifi-

cation heads, one for the end task under consideration and the other for the emotion recognition task. While training, both tasks are given equal weightage by sampling their mini-batches in equal proportions.

**Contrastive methods:** We compare our approach with Sentence-BERT/RoBERTa (**SNT**) (Reimers and Gurevych, 2019). It fine-tunes BERT and RoBERTa on natural language inference task using Siamese network architecture, with the objective to learn semantically meaningful sentence representations. Next, we compare our approach with two SCL methods: (1) **SCL-Joint** (Gunel et al., 2021): proposed for sequence classification tasks, it defines the loss function as an affine combination of CE and SCL loss. (2) **SupCLSeq** (Sedghamiz et al., 2021): a pipelined approach that first updates PLM network weights using SCL loss and dropout-based data augmentation. The updated PLM is then further fine-tuned on end tasks using CE loss. Both methods apply the CE and SCL losses only for the task under consideration, i.e., single-task setting. They do not take any explicit affective signals into account.

**Affect-aware methods:** We compare the ∗Emo models with two affect-aware PLMs: (1) **KEA** (Suresh and Ong, 2021): enriches contextualized word embeddings using valence, arousal, and dominance scores in the NRC VAD lexicon (Mohammad, 2018). The [CLS] embeddings are first used as a query vector to learn sentence embeddings in terms of attention weighted VAD-enriched contextualized embeddings. A classification head over the sentence embeddings is then used to fine-tune end tasks; (2) **SentiX** (Zhou et al., 2020): learns sentiment-aware BERT from scratch using large-scale review datasets. In addition to MLM and NSP, it adds additional pre-training objectives at token and sentence level using emoticons, sentiment lexicons, and review ratings.

### 5.2 Evaluating Emotion-awareness

The question we posed to evaluate language models for their emotion content is: Do text fragments that evoke the same emotion have similar sentence embeddings? In other words, are fragments with similar emotion content clustered together in the embedding space? Our retrofitting corpus (i.e., unseen test set in go_emotions) provides the required test bed for this study. We perform K-means clustering (#means = 28), considering [CLS] embeddings

| Representations | Clustering Metrics | | | Retrieval Metrics | | | $\Delta_{emb} \downarrow$ |
|---|---|---|---|---|---|---|---|
| | AMI↑ | ARI↑ | FMS↑ | MRR↑ | P@1↑ | MAP@r↑ | |
| BERT | 0.011 | 0.001 | 0.076 | 0.339 | 0.193 | 0.043 | 0 |
| SNT$_{BERT}$ | 0.079 | 0.011 | 0.089 | 0.414 | 0.264 | 0.06 | 0.209 |
| SentiX[†] | 0.041 | 0.006 | 0.088 | 0.355 | 0.207 | 0.052 | 1.211 |
| TLearn$_{BERT}$ | 0.378 | 0.133 | 0.229 | 0.625 | 0.503 | 0.28 | 0.941 |
| BERTEmo | 0.299 | 0.102 | 0.194 | 0.584 | 0.446 | 0.189 | 0.055 |
| RoBERTa | 0.05 | 0.014 | 0.101 | 0.374 | 0.228 | 0.051 | 0 |
| SNT$_{RoBERTa}$ | 0.117 | 0.021 | 0.099 | 0.436 | 0.289 | 0.065 | 0.731 |
| TLearn$_{RoBERTa}$ | 0.402 | 0.158 | 0.255 | 0.635 | 0.514 | 0.305 | 0.945 |
| RoBERTaEmo | 0.377 | 0.142 | 0.237 | 0.614 | 0.488 | 0.257 | 0.172 |

Table 3: Evaluating sentence embeddings for emotion content using clustering and retrieval metrics (↑: higher values are better). The last column $\Delta_{emb}$ reports the mean cosine distance between sentence embeddings obtained from emotion-aware models and their pre-trained versions. To ensure that the linguistic knowledge already present in PLMs is not inadvertently perturbed, this value should be as low as possible. BERTEmo and RoBERTaEmo are not only emotion-aware (high values for clustering and retrieval metrics) but also preserve the topology of sentence embeddings space (low values for $\Delta_{emb}$). While the compared methods take emotion aspects into account, they are not good at vector space preservation (high values for $\Delta_{emb}$). {SentiX[†]: applicable only for BERT}

of text fragments as features. Since the emotion labels are available, we apply external cluster validity indices such as adjusted mutual information (AMI), adjusted rand index (ARI), and Fowlkes Mallows score (FMS) (refer to Scikit-learn user guide) to measure clustering quality. In addition to clustering, we also consider retrieval-based metrics[3] such as mean reciprocal rank (MRR), precision@1 (P@1), and mean average precision at r (MAP@r). These metrics directly probe the nearest neighbors of given text fragments for their consistency in emotion labeling. While retrofitting PLMs for emotions, we want to preserve the lexical, syntactic, and semantic knowledge already contained in them. To quantify this property, we compute the mean cosine distance between sentence embeddings obtained from emotion-aware language models and their pre-trained counterparts (referred to as $\Delta_{emb}$; lower values are better).

We investigate[4] *Emo, SNT, TLearn, SentiX, and their pre-trained counterparts. As shown in Table 3, the BERT and RoBERTa baselines have extremely low scores across all metrics. This is because emotion aspects have not been explicitly taken into account during their pre-training phase. The sentence embeddings in SNT slightly improve the clustering and retrieval metrics. Sur-

prisingly, even though SentiX is trained on sentiment data, which has some affective aspects, it is not good at capturing emotion aspects. The straightforward way of incorporating emotions into PLMs by fine-tuning them on emotion recognition task, i.e., TLearn, drastically improves the clustering and retrieval metrics. However, it excessively alters the topology of sentence embedding space (very high $\Delta_{emb}$) and may end up overfitting the embeddings for emotions. The *Emo models retrofitted by our method are not only emotion-aware (high values for clustering and retrieval metrics) but also preserve the topology of the embedding space (low values for $\Delta_{emb}$).

Appendix D shows 2-dim UMAP plots for text fragments in the go_emotions test set. The fragments from all emotion categories are completely interleaved for BERT and RoBERTa. On the other hand, BERTEmo and RoBERTaEmo provide a good separation between different emotion categories. The cosine distances computed using BERTEmo are well calibrated for emotion content, as evident from the exemplar pairs in Table 1.

## 5.3 Evaluation on Downstream Tasks

We evaluate our method on two affective downstream tasks: (1) Sentiment analysis on Stanford sentiment treebank (sentence level) with both the graded (**SST5**) and binary (**SST2**) variants; and SemEval 2017 task 4A (**SE**) containing tweet messages; (2) Sarcasm detection using Mustard++ dataset (**Mus**) that contains sit-com utterances.

---

[3]refer to Appendix C for details on metrics.

[4]Not applicable for remaining methods: MTL and KEA include emotion signals only during end task fine-tuning; SCL-Joint and SupCLSeq do not consider emotion signals at all.

| Task | Dataset | #class | size | #token | length | Type | Source |
|---|---|---|---|---|---|---|---|
| Sentiment analysis | SST5 | 5 | 11855 | 199120 | 25±11 | sentence | (Socher et al., 2013) |
| | SST2 | 2 | 9613 | 162783 | 25±11 | sentence | (Socher et al., 2013) |
| | SE | 3 | 61854 | 1174626 | 25±7 | tweet | (Rosenthal et al., 2017) |
| Sarcasm detection | Mus | 2 | 1202 | 14219 | 15±8 | utterance | (Ray et al., 2022) |

Table 4: Dataset statistics for downstream tasks: (1) Sentiment analysis on SST5, SST2 and SemEval 2017 task 4a (SE) datasets; (2) Sarcasm detection on Mustard++ (Mus) dataset {**length**= #tokens per instance}

| Models | SST5 | SST2 | SE | Mus |
|---|---|---|---|---|
| BERT | 0.342 | 0.706 | 0.51 | 0.508 |
| BERTEmo | 0.443 | 0.854 | 0.63 | 0.55 |
| RoBERTa | 0.366 | 0.724 | 0.535 | 0.571 |
| RoBERTaEmo | 0.498 | 0.881 | 0.653 | 0.558 |

Table 5: Micro-F1 scores for KNN classification using sentence embeddings as features: ∗Emo models perform significantly better than pre-trained baselines

Table 4 details the statistics of these datasets.

While we showed the intrinsic efficacy of emotion-aware sentence representations in section 5.2, are they effective for downstream tasks? To study this, we learn KNN classifier[5] for end tasks, treating [CLS] embeddings as features. Being emotion-aware, the retrofitted ∗Emo models achieve significantly better results ($\approx 10\%$ improvements in F1-score) than baselines on both tasks (refer to Table 5). This suggests that incorporating emotion content into PLMs is beneficial for end tasks that are affective in nature. Next, we perform fine-tuning experiments where we update all network parameters during training.

Table 6 reports the micro F1-scores (averaged over 30 runs) on the downstream tasks for the fine-tuning experiments. The standard fine-tuning of BERT and RoBERTa using CE loss seems to be a hard baseline to beat for both tasks. The sentence embeddings in SNT improves it slightly for the sentiment analysis task. Though ubiquitous, the CE loss sometimes leads to poor generalization (Liu et al., 2016) and may not be robust to noisy labels (Zhang and Sabuncu, 2018). Therefore it has recently been supplemented with the SCL loss either in a joint or pipeline architecture. Though the joint learning approach in SCL-Joint could not perform well, we observed that the pipelined architecture in

SupCLSeq led to slightly improved F1-scores on both tasks. It should be noted that these approaches have not taken any emotion signals into account during fine-tuning.

Being pre-trained on large review datasets such as Yelp and Amazon that mainly contain sentiment signals, SentiX attains the highest F1-score on sentiment analysis task (for BERT). However, it does not generalize to other tasks (e.g., inferior results on sarcasm detection). The knowledge (valence, arousal, dominance) embedded approach in KEA unexpectedly does not perform well on any task. The PLMs retrofitted for emotions using transfer learning (TLearn) and multi-task learning (MTL) perform better than the pre-trained baselines on both tasks, exhibiting the positive impact the external emotion signals provide. In this work, we seek to learn emotion-aware PLMs, but not at the expense of displacing existing linguistic knowledge (by overfitting them for emotions). While the VSP loss helps in preserving the topology of the sentence embedding space, the contrastive loss brings robustness to our method. This helps ∗Emo models achieve better results than other approaches, with $\approx 1\%$ statistically significant improvements ($p\text{-}val$ $< 0.01$ for one-tailed student's $t$-test with sample size 30) over pre-trained baselines.

***Comparison with ChatGPT:*** Zhang et al. (2023) have reported results on gpt-3.5-turbo with zero-shot and in-context learning for 13 sentiment analysis tasks. Table 7 compares these results with the fine-tuned RoBERTaEmo (our method). As we can see, RoBERTaEmo performs significantly better than gpt-3.5-turbo on the sentence-level sentiment classification datasets considered in this work. It will be interesting to compare RoBERTaEmo with a fine-tuned gpt-3.5-turbo. We will leave this for future work.

### 5.3.1 Ablation Study

***Vector space preservation:*** Retrofitting methods for static embeddings have consistently used a reg-

---

[5]The network weights are fixed only for the KNN classification experiments. The rest of the experiments in this section jointly update parameters of both the PLM and classification head (i.e., standard fine-tuning)

| Models | BERT | | | | RoBERTa | | | |
|---|---|---|---|---|---|---|---|---|
| | SST5 | SST2 | SE | Mus | SST5 | SST2 | SE | Mus |
| pre-trained | 0.5291 | 0.9103 | 0.6983 | 0.5935 | 0.5591 | 0.9335 | 0.7022 | 0.5708 |
| SNT | 0.5356 | 0.9139 | 0.6989 | 0.5731 | 0.5634 | 0.9368 | 0.7064 | 0.5698 |
| SupCLSeq | 0.5304 | 0.9139 | 0.6834 | 0.5987 | 0.568 | **0.9449** | 0.6975 | 0.5768 |
| SCL-Joint | 0.5074 | 0.9125 | 0.6916 | 0.5526 | 0.5572 | 0.9445 | 0.6981 | 0.5014 |
| TLearn | 0.5347 | 0.9136 | **0.702** | 0.5869 | 0.5643 | 0.9374 | **0.7078** | 0.5734 |
| MTL | 0.5294 | 0.9104 | 0.6884 | **0.6041** | **0.5684** | 0.9407 | 0.7006 | **0.5942** |
| SentiX† | 0.5471 | 0.9186 | 0.6934 | 0.5638 | - | - | - | - |
| KEA | 0.5316 | 0.9060 | 0.6977 | 0.5913 | 0.5492 | 0.9368 | 0.7074 | 0.5828 |
| *Emo | **0.5364** | **0.9141** | 0.7026 | 0.6119 | 0.5688 | 0.9454 | 0.7097 | 0.614 |
| *Emo$_{\lambda=0}$ | 0.5249 | 0.9092 | 0.7010 | 0.5987 | 0.5655 | 0.8417 | 0.7090 | 0.5670 |

Table 6: Micro F1-Scores for fine-tuning experiments on sentiment analysis (SST5, SST2, SemEval 2017 task 4a (SE)) and sarcasm detection (Mus): compares *Emo with **pre-trained** baselines in BERT, RoBERTa, and other approaches (**Bold+Underline**: highest; **Bold**: next highest). **Top block:** Sentence-BERT/RoBERTa (SNT) and SCL methods (SupCLSeq and SCL-Joint) do not take explicit emotion signals into account; **Middle block:** transfer learning (TLearn), multi-task learning (MTL), SentiX, and KEA take affective signals into account and are emotion-aware; **Last block:** emotion-aware models learned by our method. *Emo statistically significantly better than pre-trained versions with $p\text{-}val < 0.01$ as computed using one-tailed student's $t$-test (sample size=30). {SentiX†: applicable only for BERT}

| Model | SST5 | SST2 | SE |
|---|---|---|---|
| ChatGPT$_{zero-shot}$ | 0.48 | 0.9360 | 0.6940 |
| ChatGPT$_{few-shot}$ | 0.5187 | 0.9527 | 0.6647 |
| RoBERTaEmo | 0.5688 | 0.9454 | 0.7097 |

Table 7: Micro-F1 scores on sentiment analysis. The fine-tuned RoBERTaEmo performs better than Chat-GPT in both zero-shot and few-shot in-context learning

| Proj | SST5 | SST2 | SE | Mus |
|---|---|---|---|---|
| I | 0.5677 | 0.9425 | 0.7055 | 0.6059 |
| Linear | 0.5694 | 0.943 | 0.7105 | 0.5964 |
| MLP | 0.5688 | 0.9454 | 0.7097 | 0.614 |

Table 8: Micro-F1 scores on end tasks. RoBERTaEmo projection head varied as: **I:** no projection; **Linear:** [CLS] projected to $\mathbb{R}^{64}$; **MLP:** [CLS] projected to $\mathbb{R}^{64}$ with a non-linear hidden layer of size 768

ularization term to preserve the topology of pre-trained vector space. However, such a term has not been considered in the context of PLMs. By constraining sentence representations to be closer to their pre-trained version, The $L_{vsp}$ term in our loss function guides weight updates for emotions such that the linguistic knowledge already present in PLMs is not distorted. When we retrofitted BERT and RoBERTa without vector space preservation (*Emo$_{\lambda=0}$ in Table 6), the performance on both the end tasks consistently deteriorated.

***On projection head:*** The SCL methods for text

(Gunel et al., 2021; Sedghamiz et al., 2021) apply contrastive loss directly on the encoder output, i.e., [CLS] embeddings (or Identity (**I**) projection head). However, for images, as suggested by Chen et al. (2020), applying contrastive loss directly on the encoder (ResNet) output inadvertently results in a loss of information that may be useful for downstream tasks. To avoid such unintended effects, we first map encoder output to a 64-dim vector space using two projection head variants, i.e., Linear and MLP, as described in section 4. Table 8 shows the F1-scores on end tasks with RoBERTaEmo variants that are learned using different projection heads. The results indicate that the Linear and MLP projection heads perform better than directly using the encoder output[6].

### 5.3.2 Few-shot Learning Experiments

We perform few-shot learning experiments on the sentiment analysis task. For all datasets, we first sample train data of various sizes from the original set such that the #training examples are in [20, 50, 100, 500], keeping the original label distribution intact. We then fine-tune BERTEmo and BERT on these data sizes and compare their micro-F1 scores on the original test set. As shown in Figure 2, the emotion-aware BERTEmo outperforms its pre-trained counterpart BERT across

---

[6]The MLP head is more robust. The variance in F1-scores across multiple runs is relatively lower with MLP than Linear.

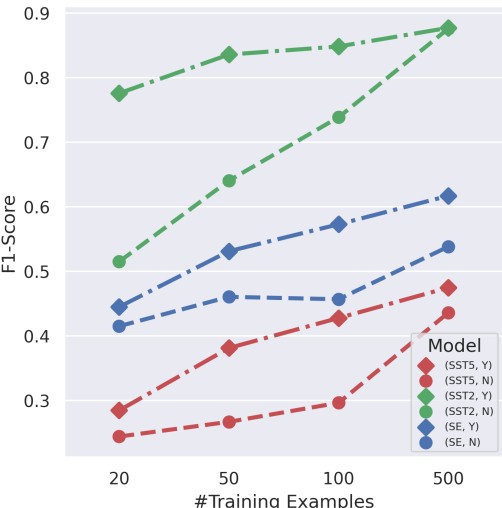

Figure 2: Few-shot learning experiments on the sentiment analysis task: BERTEmo (♦**Y**) performs significantly better than BERT (●**N**)

all datasets by a significant margin (similar observation for RoBERTaEmo; refer to Figure 3 in Appendix B). The difference in performance gradually decreases with an increase in #training examples. This suggests that the affective knowledge on emotions, as captured by our retrofitting method, helps improve end tasks, especially in few-shot learning scenario.

## 6   Conclusion and Future Work

We present a novel retrofitting method to learn emotion-aware PLMs in a contrastive learning setting using emotion labeling in the go_emotions (Demszky et al., 2020) corpus. It updates PLM network weights such that the text fragments exhibiting similar emotions are encoded nearby in the representation space, and fragments with different emotions are pushed apart while preserving the linguistic knowledge originally captured during the PLM pre-training phase. The emotion-aware models (∗Emo) learned by our method perform better than their pre-trained counterparts (about 1% improvement in F1-score) and other benchmarks, with significant gains in few-shot learning setting.

Going forward, we want to extend our retrofitting method to other affective resources such as the NRC VAD lexicon, emoticons, etc. We also plan to investigate affective content in large language models such as GPT4.

## Limitations

In this work, we have used the go_emotions dataset primarily due to its large size. The dataset has been created from English Reddit comments. Being retrofitted solely on go_emotions, ∗Emo may contain Reddit-specific nuances. It will be interesting to learn jointly from multiple emotion recognition datasets spanning different genres and domains such as news, tweets, blogs, health, politics, etc. However, this may also bring in additional complexities. For instance, we do not know which domain or genre should be given more importance, how to handle datasets that vary a lot in size, etc.

We demonstrated the effectiveness of ∗Emo models on two downstream tasks. We believe these models can generalize to other affective tasks such as hate speech detection, bias detection, and empathetic agents. However, it is difficult to comment on their effectiveness on general NLP tasks such as entity extraction, grammatical error correction, etc. Though the vector space preservation term in our loss function keeps a check on the PLM network weights so that the existing linguistic knowledge is not inadvertently perturbed, a few changes are bound to happen to accommodate the emotion aspects. It will be interesting to compare ∗Emo with their pre-trained counterparts on general benchmarks such as GLUE and SuperGLUE.

## Ethics Statement

The go_emotions dataset has been created from English Reddit comments which are known to contain toxic/offensive language. These comments are also demographically biased toward young male users. Though the creators of go_emotions have taken extensive care in data filtering, pre-processing, and masking steps to address the bias and offense-related issues, the dataset might still inadvertently contain inappropriate content. This might then flow directly into ∗Emo models, degrading their quality.

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

## A  Training Details

In this section, we provide the complete hyper-parameter grid search details and the best combinations selected thereof. As described in section 4, we treat the PLMs in BERT and RoBERTa as encoding function $\text{Enc}(\cdot)$. The $[\text{CLS}]$ embeddings obtained from the $\text{Enc}(\cdot)$ function are then passed to a projection head $\text{Proj}(.)$. We consider three variants for $\text{Proj}(.)$: {Identity, Linear, MLP}. For both $\text{Enc}(.)$ and $\text{Proj}(.)$, we set weight-decay to 0.01 and dropout to 0.1. The temperature factor $\tau$ in NT-Xent loss in eq. 1 is varied as {0.05, 0.1, 0.2}. We set the batch size B to 64, with cross-batch memory in XBM (Wang et al., 2020) varied as {512, 1024, 2048}. The regularizer for vector space preservation loss, $\lambda$, is varied as {0.01, 0.05, 0.1, 0.5}. The hyper-parameters relevant for training are: ReLU activation; AdamW (Loshchilov and Hutter, 2019) optimizer with Cosine decay and warm-up (3 steps); learning rate varied as {5e-06, 1e-05}; and 30 epochs with early stopping using official validation set. The training batches are generated using two sampler variants: MPerClassSampler and StratifiedSampler.

For experimentation, we used Nvidia DGX A100 GPUs with a memory size of 20GB RAM. Each configuration, on average, took 17 minutes to run.

| hyperparameter | BERTEmo | RoBERTaEmo |
|---|---|---|
| sampler | MPerClass | MPerClass |
| $\lambda$ | 0.05 | 0.01 |
| temperature $\tau$ | 0.05 | 0.1 |
| $\text{Proj}(\cdot)$ | Linear | MLP |
| batch size | 64 | 64 |
| memory size | 1024 | 512 |
| learning rate | 5e-06 | 5e-06 |

Table 9: The best hyper-parameter configurations for emotion-aware PLMs: BERTEmo and RoBERTaEmo

We find the best hyper-parameter configuration setting in the following way. Post training, for

each combination, we compute two metrics: (1) the clustering metric in **AMI** to measure the emotion-awareness of the retrofitted PLM; (2) the mean cosine distance between sentence embeddings obtained from the retrofitted PLM and its pre-trained version, i.e., $\mathbf{\Delta_{emb}}$, to measure the vector space preservation quality. We first filter configurations for which $\Delta_{emb}$ is less than 0.1 for BERT and 0.2 for RoBERTa. We then choose the configuration with the highest AMI from the filtered set. We consider the performance on the SST5 dataset (sentiment analysis task) for tie-breaking. Table 9 reports the best hyper-parameter configurations for BERTEmo and RoBERTaEmo. To implement our retrofitting method, we used Pytorch Metric Learning library (Musgrave et al., 2020b).

## B Training Details: Downstream tasks

For downstream tasks, we add a linear classification head over the sentence embeddings in [CLS] and jointly fine-tune both the PLM network weights and the classification head using cross-entropy loss. We implement this using Auto Classes[7] from the huggingface library. The relevant hyper-parameters are: AdamW optimizer with learning rate in {1e-05, 2e-05, 3e-05}, dropout=0.1, and batch size=32.

While section 5.3.2 compares BERT with the emotion-aware BERTEmo on few-shot learning experiments for the sentiment analysis task, here we report results for RoBERTa and its emotion-aware version in RoBERTaEmo. As can be seen from Figure 3, RoBERTaEmo performs significantly better than its pre-trained version RoBERTa across all datasets in the limited data setting.

## C Evaluating Emotion-awareness: Metrics

As described in section 5.2, we use clustering and retrieval metrics for evaluating emotion awareness. The existing labeling of text fragments in the go_emotions test set provides us with *true* clustering. Whereas the clustering induced by the K-means algorithm gives the *predicted* clustering. The partition of text fragments provided by the *true* and *predicted* clustering is then used to compute the clustering validity indices such as adjusted mutual information (AMI), adjusted rand index (ARI), and Fowlkes Mallows score (FMS) (refer to Scikit-learn user guide). Unlike clustering, the retrieval-based metrics directly probe the neighborhood of

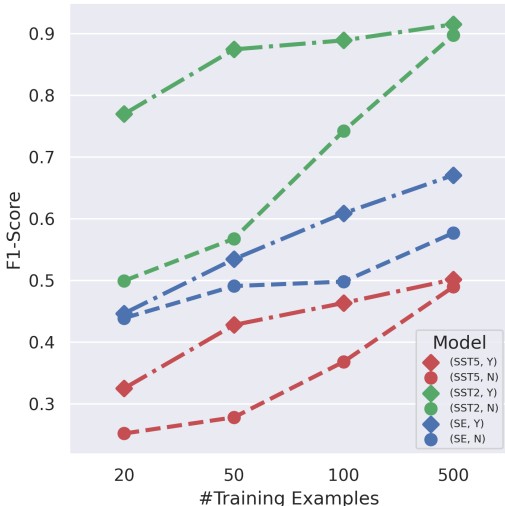

Figure 3: Few-shot learning experiments with the sentiment analysis task: RoBERTaEmo (♦Y) performs significantly better than RoBERTa (●N)

text fragments for their consistency in emotion labeling. The precision@1 metric checks the immediate neighbor of the queried text fragment. The mean reciprocal rank (MRR) metric considers the reciprocal rank of the closest fragment with the same emotion label as the query fragment. The mean average precision at r (MAP@r) metric considers average precision till rank r, where r is the number of text fragments with the same emotion label as the query fragment. Refer (Musgrave et al., 2020a) for details.

## D Evaluating Emotion-awareness: UMAP plots

We visualize 2-dim UMAP (McInnes et al., 2018) plots for text fragments in the go_emotions test set, comparing ∗Emo with their pre-trained counterparts. We learn the 2-dim embeddings using the umap-learn library[8] with the following hyper-parameter setting: #neighbors=15; min_dist=0.1; distance metric=Euclidean. As we can see from Figure 4 and Figure 5, The text fragments from all emotion categories are completely interleaved for BERT and RoBERTa. On the other hand, BERTEmo and RoBERTaEmo provide a good separation between different emotion categories.

---

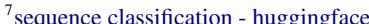

[7]sequence classification - huggingface

[8]https://pypi.org/project/umap-learn/

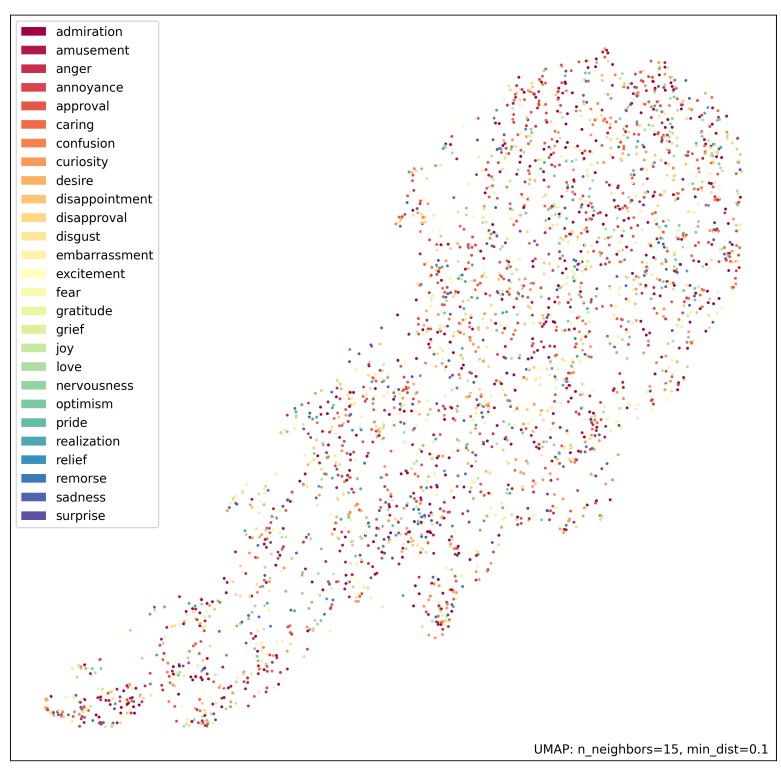

(a) BERT [CLS] embeddings

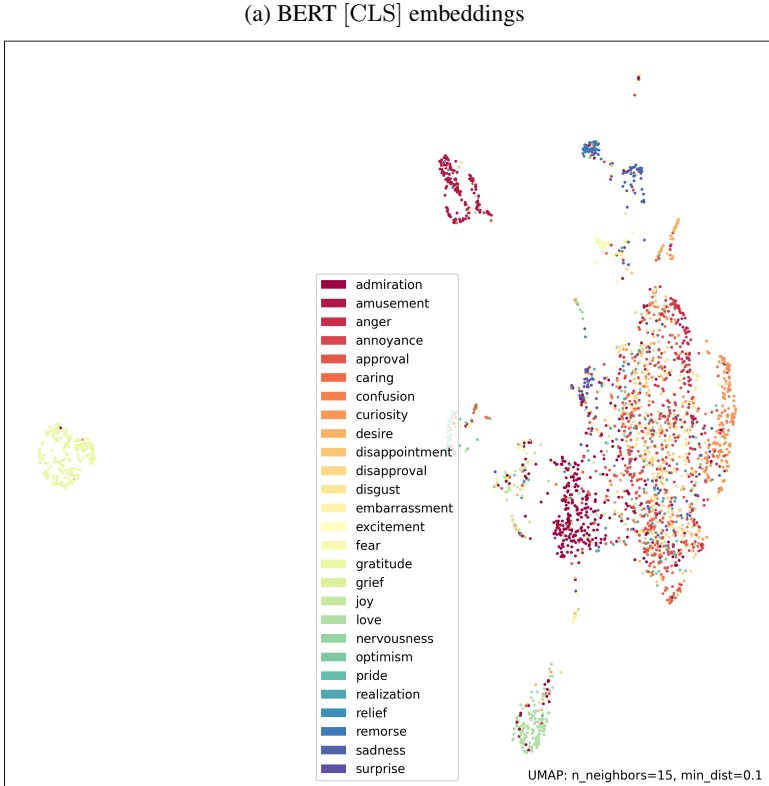

(b) BERTEmo [CLS] embeddings

Figure 4: 2-dim UMAP plots for text fragments in the go_emotions test set: compares BERT with BERTEmo

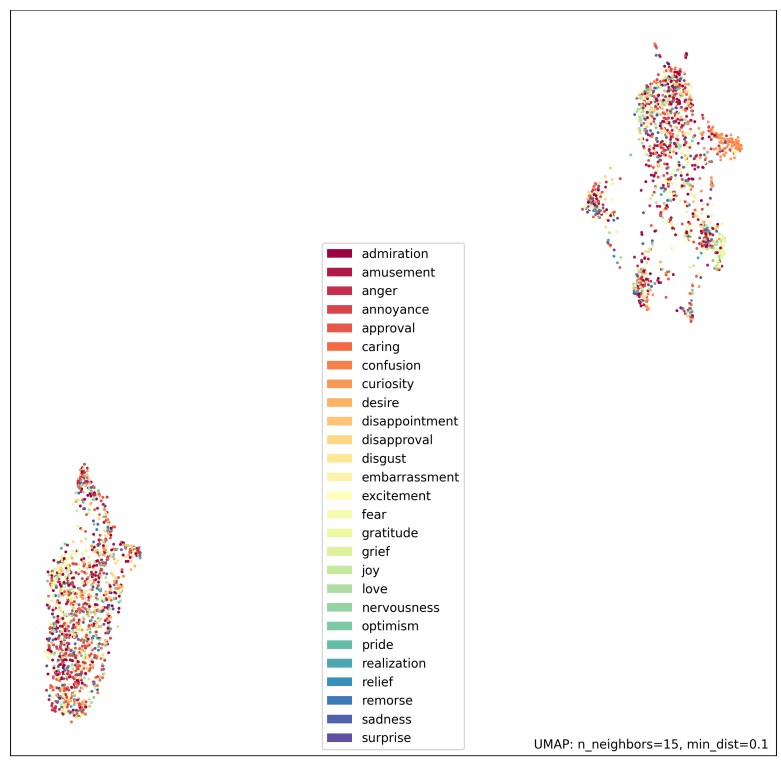

(a) RoBERTa [CLS] embeddings

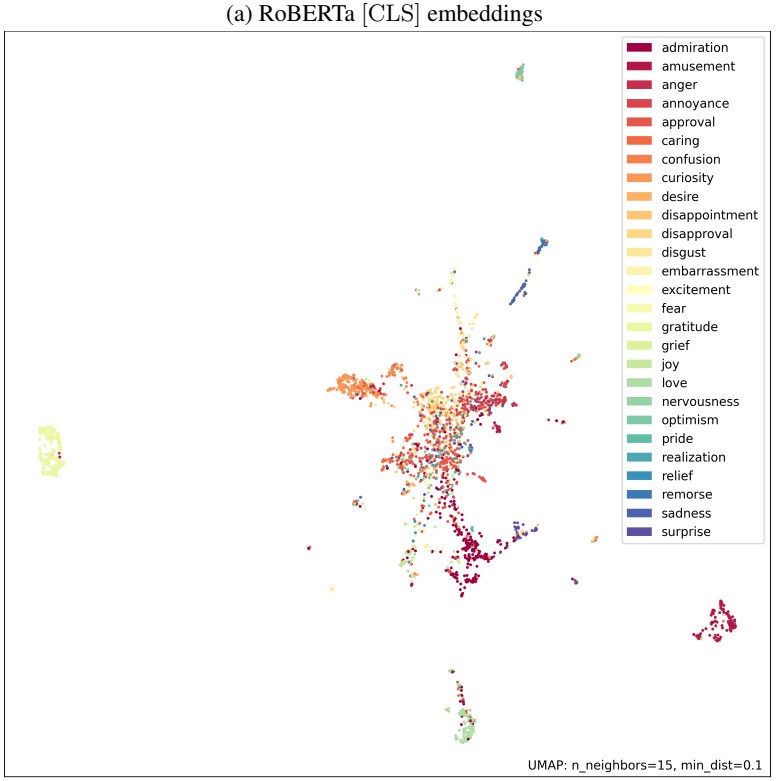

(b) RoBERTaEmo [CLS] embeddings

Figure 5: 2-dim UMAP plots for text fragments in the go_emotions test set: compares RoBERTa with RoBERTaEmo