# OpenReview forum: "Retrofitting Light-weight Language Models for Emotions using Supervised Contrastive Learning"
_EMNLP/2023/Conference — EMNLP 2023 Main_

### Official Review · Reviewer_HtpC · 2023-08-05

**Soundness:** 4

**Excitement:**

3: Ambivalent: It has merits (e.g., it reports state-of-the-art results, the idea is nice), but there are key weaknesses (e.g., it describes incremental work), and it can significantly benefit from another round of revision. However, I won't object to accepting it if my co-reviewers champion it.

**Paper Topic And Main Contributions:**

This paper presents a novel retrofitting method to induce emotion aspects into pre-trained language models, which updates pre-trained network weights using contrastive learning.

**Reasons To Accept:**

1. The authors conduct sufficient experiments to verify the effectiveness of their method.

2. This paper is clearly organized and well-written. The description of the method is easy to understand.

**Reasons To Reject:**

1. The method in this paper is limited in innovation.

2. The authors did not compare their method with the latest SOTA.

3. The papers cited by the authors in the related work section lacks the most recent papers (there is no paper for 2023).

4. I don't think it is necessary to spend too much space on the description of some existing work, such as subsection 5.1 and Table 4. Instead, the space should be devoted to the analysis of experimental results.

**Reproducibility:**

3: Could reproduce the results with some difficulty. The settings of parameters are underspecified or subjectively determined; the training/evaluation data are not widely available.

**Reviewer Confidence:**

4: Quite sure. I tried to check the important points carefully. It's unlikely, though conceivable, that I missed something that should affect my ratings.

---

> ### Author Rebuttal · Authors · 2023-08-27
>
> ###### We would like to start by thanking you for your positive comments on our experimentations and well-written content. We have carefully considered your comments in the "Reasons to Reject" section and have provided point-wise responses below.
>
> ### On Innovation:
> ###### In NLP, supervised contrastive learning (SCL) has been primarily used in a task-specific setting (either in a pipelined or joint-learning fashion). It has not been explored as a representation learning tool. To the best of our knowledge, our work is the first to apply SCL as a representation learning tool for retrofitting language representations.
> ###### In retrofitting setting, the SCL loss (NT-Xent) needs to be appropriately regularized to prevent the inadvertent perturbation of linguistic knowledge (semantic relations, concept similarities, etc.) already present in PLMs. The vector space preservation loss introduced in this work provides this much-needed regularization. As reported in Table 6 (last row), an emotion-aware model learned using only the SCL loss does not perform well on downstream tasks (performance comparable to the pre-trained baseline, or even less in some cases). On the other hand, including the vector space preservation loss results in a robust emotion-aware model that outperforms pre-trained baseline as well as other compared work.
>
> ### Comparison with latest SOTA:
> ###### We experimented with gpt-3.5-turbo (model behind ChatGPT) on the SST5 dataset (sentiment analysis task) in a zero-shot setting. As of 9th June, we obtained an F1-score of 0.498 vs. 0.529 for the pre-trained BERT baseline (mentioned on line no 576). We are currently evaluating it on the remaining datasets. In fact, a recent paper [1] has reported results for zero-shot and few-shot** experiments with gpt-3.5-turbo on the same sentiment analysis dataset we have considered. The RoBERTaEmo (our method) has better accuracy than both the zero-shot and the few-shot gpt-3.5-turbo. The following table shows the comparison.
>
> 							SST5	SST2	SE(SemEval-Twitter)
> 	RoBERTaEmo (ours)	0.5688	0.9454	0.7097
> 	ChatGPT[1]		0.48	0.9360	0.6940
> 	ChatGPT_few-shot[1]	0.5187	0.9527	0.6647
> We will include these findings in the revised version.
> ###### ** The few-shot results on gpt-3.5-turbo vary across 1-shot, 5-shot, and 10-shots with no indication of improved results with increase in #few-shots.
>
> ### Related Work - no paper from 2023:
> ###### We have systematically scanned all 2023 papers on ACL Anthology. The following table shows the #publications containing the word 'sentiment' in the paper title across ACL venues (2023).
> 			ACL+Findings	EACL	SemEval		starsem	TACL	OtherWorkshops/Events
> 	#papers	66		9	29		1	1	38
> ###### For sentiment analysis (SA), the research community has been mainly focussing on the following sub-topics: Multi-modal SA, Aspect-based SA, User-Product-based SA, SA for low-resource/language setting, and African SA (due to SemEval 23 shared task). However, in the current work, our focus is on the standard multi-class sentiment classification task. This is relatively less explored at ACL venues (2023), because of which we could only find one related 2023 paper from the anthology [2]. However, that paper is also  contextually not relevant to our work as it mainly focuses on designing prompt templates. We will explore other non-ACL venues and include a comparison if we find a suitable related work.
>
> ### On space utilization (comment that the paper spends too much space on subsection 5.1, Table 4 ...):
> ###### We will look at this again to see what we can improve. We will consider moving Table 4 to Appendix given that the datasets for sentiment analysis and sarcasm detection tasks are widely known.
>
> ### On Reproducibility:
> ###### 1. Hyper-parameter setting: We have provided the complete hyper-parameter grid search details, including network architecture, loss functions, and optimizer parameters, in the Appendix A (including the hardware details and runtimes). It also reports the best hyper-parameter configurations. To the best of our knowledge, we have discussed all relevant parameter details. However, if the details are inadequate on some of the parameters, we will be happy to expand.
> ###### 2. Training/evaluation data: The corpus used for retrofitting (i.e., go_emotions) and the datasets used for downstream tasks are widely available (on author websites, huggingface/datasets). We have cited the respective papers and used the author suggested train/validation/test splits.
> ###### 3. The micro-F1 scores reported in Table 6 are averaged over 30 runs. The emotion-aware models retrofitted by our method perform better than pre-trained BERT and RoBERTa with statistical significance (p-value < 0.01) as computed using one-tailed student's t-test (sample size = 30) (Line no 502).
>
> ### On Soundness:
> ###### The claims/arguments in our paper are supported by extensive experimentation, including ablation study (Sec. 5.3.1) and few-shot learning experiments (Sec. 5.3.2). The retrofitting method in Section 4 provides a clear explanation of the training data generation, neural network architecture, loss functions, and regularizer. It would be helpful if the reviewer could point to any specific technical or methodological problems with our work.
>
> ####
> ####
> ###### In view of the point-wise responses mentioned above, we respectfully request that the reviewer consider increasing the score.
> ###### Lastly, thank you for your valuable feedback. We appreciate your suggestions, as they will help us improve our work.
>
> ### References:
> ###### [1] Zhang, W., Deng, Y., Liu, B., Pan, S.J., & Bing, L. (2023). Sentiment Analysis in the Era of Large Language Models: A Reality Check. ArXiv, abs/2305.15005.
> ###### [2] A Simple Yet Strong Domain-Agnostic De-bias Method for Zero-Shot Sentiment Classification (Zhao et al., Findings ACL 2023)

---

### Official Review · Reviewer_tXoc · 2023-08-07

**Typos Grammar Style And Presentation Improvements:** Nothing major bug a few typos and edi…
**Soundness:** 4

**Excitement:**

4: Strong: This paper deepens the understanding of some phenomenon or lowers the barriers to an existing research direction.

**Missing References:**

Nothing major missing perhaps approaches not involving PLMs.

**Paper Topic And Main Contributions:**

The paper proposes retrofitting to improve affect aware representation of well established PLMs. Multiple experiments are performed to support the improvement seen in PLM through the proposed retrofitting.

**Questions For The Authors:**

Do you invision the approach scaling across to other languages? Or, even non standard English?
Is the approach really worth it given the performance discrepancy evident between newer generative models and PLMs?

What about other loss functions outside of contrastive? I do feel the choice of loss functions could be better justified.

**Reasons To Accept:**

Openly available and computationally less expensive PLMs can be retrofitted to outperform other established approaches to making the models more affect aware. The experiments are extensive, valid, and well performed.

**Reasons To Reject:**

The only potential reason would the lacking comparison between these retrofitted models and newer models e.g. GPTs.


**Reproducibility:**

4: Could mostly reproduce the results, but there may be some variation because of sample variance or minor variations in their interpretation of the protocol or method.

**Reviewer Confidence:**

4: Quite sure. I tried to check the important points carefully. It's unlikely, though conceivable, that I missed something that should affect my ratings.

---

> ### Author Rebuttal · Authors · 2023-08-27
>
> Thank you for your valuable feedback. It really helps us improve our work.
>
> ### On Scaling:
> ###### 1. To non-standard English: Yes, it seems to scale. In fact, one of the sentiment analysis datasets (SemEval 2017 Task 4A) we have experimented with has tweets as input. Oftentimes, tweets have non-grammatical sentences and spelling mistakes and variations.
> ###### 2. To other languages: There could be multiple threads to explore here: (a) If the target language has a sizeable retrofitting corpus, the proposed approach can be directly applied. (b) Considering English as a source language, retrofitting corpus for the target language can be obtained using an off-the-shelf Machine Translation (MT) system. The quality of the generated corpus will depend on how accurate the MT model is for the given <source, target> language pair. Even when it is reasonably accurate, MT itself may bring in noise. (c) It is worth checking the applicability of multilingual BERT/RoBERTa (e.g., bert-base-multilingual from Google Research). Does a multilingual PLM, retrofitted for emotions using English, accurately scale to other languages?
> ###### Thank you for pointing us to the language aspect. It opens up a new line of exploration for us.
>
> ### On the choice of loss function:
> ###### We wanted to retrofit PLMs using a corpus that provides sentence-level emotion annotations, i.e., input: sequence; output: label. We had initially considered two loss functions: cross-entropy loss and contrastive loss.
> ###### We chose contrastive loss for our method for two reasons. First, cross-entropy loss is not robust to noisy data [1] (it may perform poorly if the training data has errors/inconsistencies). Second, cross-entropy loss sometimes does not generalize well [2] due to learning poor margins. Contrastive loss is more robust to noisy data and generalizes better than cross-entropy loss. It focuses on learning the relative distances between different emotion categories rather than the absolute probabilities of each category, making it less sensitive to errors in the training data. We will explain this in more detail in the revised version.
>
> ### Comment on newer generative models:
> ###### We experimented with gpt-3.5-turbo (model behind ChatGPT) on the SST5 dataset (sentiment analysis task) in a zero-shot setting. As of 9th June, we obtained an F1-score of 0.498 vs. 0.529 for the pre-trained BERT baseline (mentioned on Line no 576). We are currently evaluating it on the remaining datasets. In fact, a recent paper [3] has reported results for zero-shot and few-shot experiments with gpt-3.5-turbo on the same sentiment analysis dataset we have considered. The RoBERTaEmo (our method) has better accuracy than both the zero-shot and the few-shot gpt-3.5-turbo. The following table shows the comparison.
>
> 							SST5	SST2	SE(SemEval-Twitter)
> 	RoBERTaEmo (ours)	0.5688	0.9454	0.7097
> 	ChatGPT[3]		0.48	0.9360	0.6940
> 	ChatGPT_few-shot[3]	0.5187	0.9527	0.6647
> ###### We will include these findings in the revised version. With each passing day, we are witnessing new developments in generative LLMs. Future versions might fare better on affective end-tasks. However, the method proposed in this work is generic and can also be applied to LLMs.
>
> ####
> ####
> ###### Lastly, thank you for your comments on our extensive experimentation. The availability of open source and computationally less expensive PLMs are essential for many domain-specific applications. Thank you for acknowledging our efforts in building such PLMs for the human affects domain.
> ### References:
> ###### [1] Zhilu Zhang and Mert Sabuncu. Generalized cross-entropy loss for training deep neural networks with noisy labels. In Advances in neural information processing systems, pages 8778–8788, 2018.
> ###### [2] Gamaleldin Elsayed, Dilip Krishnan, Hossein Mobahi, Kevin Regan, and Samy Bengio. Large margin deep networks for classification. In Advances in neural information processing systems, pages 842–852, 2018.
> ###### [3] Zhang, W., Deng, Y., Liu, B., Pan, S.J., & Bing, L. (2023). Sentiment Analysis in the Era of Large Language Models: A Reality Check. ArXiv, abs/2305.15005.

---

### Official Review · Reviewer_MQD5 · 2023-08-07

**Soundness:** 4

**Excitement:**

3: Ambivalent: It has merits (e.g., it reports state-of-the-art results, the idea is nice), but there are key weaknesses (e.g., it describes incremental work), and it can significantly benefit from another round of revision. However, I won't object to accepting it if my co-reviewers champion it.

**Paper Topic And Main Contributions:**

This paper proposes a retrofitting method with supervised contrastive learning to retrofit the PLM into an emotional-aware one. The retrofitting objective has two parts: a supervised contrastive loss and a vector space preservation loss. This paper post-processes the PLMs (BERT and RoBERTa) in an emotional corpus utilizing the retrofitting objective and then fine-tuning the new PLM on downstream tasks. The experiment result shows that this method has about a 1% improvement in F1-score compared with baseline methods.

**Questions For The Authors:**

Question A: As you mentioned in Line 331, the classifier, as well as the PLM, are jointly optimized. On the other hand, you use the KNN classifier (which means the PLMs are fixed since [CLS] embeddings are regarded as features?) described in Line 454. What is the correct one?

Question B: In classification tasks, a simple yet strong baseline is adding a classifier with a single softmax layer and optimising the parameters from the classifier and PLM. Upon this method, the cosine distance between different instances is not as important as claimed in this paper because this method will map the [CLS] embeddings into another space (the output distribution). Thus, how to prove the importance of the cosine distance of [CLS] embeddings?

**Reasons To Accept:**

1. Orthogonal methods for a specific domain are welcomed.
2. Experiments are exhaustive and the results are positive on most datasets.

**Reasons To Reject:**

1. Retrofitting method is a kind of fine-tuning strategy which is not new. The loss function in this paper is also not new. Therefore the novelty is limited.
2. The performance improvement is not significant.
3. The corpus for retrofitting method is not large (compared with other large-scale unlabeled corpus). Therefore this work is more likely to be an adaptation method in the emotional domain.

**Reproducibility:**

4: Could mostly reproduce the results, but there may be some variation because of sample variance or minor variations in their interpretation of the protocol or method.

**Reviewer Confidence:**

3: Pretty sure, but there's a chance I missed something. Although I have a good feel for this area in general, I did not carefully check the paper's details, e.g., the math, experimental design, or novelty.

---

> ### Author Rebuttal · Authors · 2023-08-27
>
> Thank you for your valuable feedback. We appreciate it and will take it into account as we continue to improve our work.
> ### Question A:
> ###### The classifier, as well as the PLM, are jointly optimized (Line 331). The main results reported in Table 6 for sentiment analysis and sarcasm detection have both the classifier and the PLM jointly optimized.
> ###### With the KNN classifier (Line 454), we wanted to investigate the following: emotion-awareness of PLMs in retrieval scenarios, e.g., fetching similar/alternative sentences from data-store for empathetic agents. We investigated this by comparing pre-trained BERT/RoBERTa with their emotion-aware versions using the KNN classifier. Given a test sentence, we retrieve its k closest neighbors in the embedding space using cosine similarity and use their emotion labels to determine the test sentence's category. Our results clearly show that the emotion-aware embeddings perform much better than their pre-trained counterparts. The PLMs are fixed only for this experiment (Table 5). In this sense, the KNN experiments shall be explained in section 5.2, i.e., "evaluating emotion-awareness"
> ###### Thanks a lot for pointing this out. We will clarify this and present it better in the revised version.
>
> ### Question B:
> ###### Agree that it is reasonable to assume that a simple baseline approach of adding a classifier with a single softmax layer and optimizing the parameters from the classifier and PLM would generally work well. After a model is fine-tuned for a given task, the [CLS] embeddings would anyway become task-specific and dataset-specific. In that case, the initial cosine distances between [CLS] embeddings should not matter much post fine-tuning. But this assumption is precisely what we wanted to investigate. The question is: When you start the training, how are the sentences with different emotion categories placed in the vector space (i.e., [CLS] embeddings)? Can you eventually learn better models for affective end-tasks by apriori making the [CLS] embeddings emotion-aware? Our results (Table 6) show that this is indeed the case. The emotion-aware [CLS] embeddings indeed help learn better models for end-tasks compared to the simple baseline.
>
> ### On retrofitting data size:
> ###### As you rightly mentioned, retrofitting methods do not learn from scratch. They take an existing PLM and update it for the stated objective. This means they do not require as much training data as methods that learn PLMs from scratch. The corpus (go_emotions) that we have used for retrofitting has 58,009 sentences marked for their emotion categories. We believe that this is a sizeable corpus for retrofitting setting, as we are not learning PLMs from scratch.
>
> ### On Novelty:
> ###### In NLP, supervised contrastive learning (SCL) has been primarily used in a task-specific setting (either in a pipelined or joint-learning fashion). It has not been explored as a representation learning tool. To the best of our knowledge, our work is the first to apply SCL as a representation learning tool for retrofitting language representations.
> ###### In retrofitting setting, the SCL loss (NT-Xent) needs to be appropriately regularized to prevent the inadvertent perturbation of linguistic knowledge (semantic relations, concept similarities, etc.) already present in PLMs. The vector space preservation loss introduced in this work provides this much-needed regularization. As reported in Table 6 (last row), an emotion-aware model learned using only the SCL loss does not perform well on downstream tasks (performance comparable to the pre-trained baseline, or even less in some cases). On the other hand, including the vector space preservation loss results in a robust emotion-aware model that outperforms pre-trained baselines as well as other compared work.
>
> ### On "The performance improvement is not significant":
> ###### We experimented with gpt-3.5-turbo (model behind ChatGPT) on the SST5 dataset (sentiment analysis task) in a zero-shot setting. As of 9th June, we obtained an F1-score of 0.498 vs. 0.529 for the pre-trained BERT baseline (mentioned on Line no 576). In fact, a recent paper [1] has reported results for zero-shot and few-shot experiments with gpt-3.5-turbo (model behind ChatGPT) on the same sentiment analysis dataset we have considered. The RoBERTaEmo (our method) has better accuracy than both the zero-shot and the few-shot gpt-3.5-turbo. The following table shows the comparison.
>
> 							SST5	SST2	SE(SemEval-Twitter)
> 	RoBERTaEmo (ours)	0.5688	0.9454	0.7097
> 	ChatGPT[1]		0.48	0.9360	0.6940
> 	ChatGPT_few-shot[1]	0.5187	0.9527	0.6647
> We will include these findings in the revised version.
>
> ####
> ###### Lastly, thank you for your comment on the exhaustive experimentation and appreciating our domain-specific work on emotion-aware language models.
> ### References:
> ###### [1] Zhang, W., Deng, Y., Liu, B., Pan, S.J., & Bing, L. (2023). Sentiment Analysis in the Era of Large Language Models: A Reality Check. ArXiv, abs/2305.15005.

---

### Meta-Review · Area_Chair_RWVR · 2023-09-17

**Recommendation:** 4

**Metareview:**

The reviewers agree that this is a solid paper which contains a comprehensive evaluation. The evaluation results indicate that the proposed method works well.

The main reserve expressed by the reviewers is the limited novelty of the method. However, the authors' answer emphasise the fact that even though supervised contrastive learning has been used before, this is the first time when it is used as a representation learning tool. In response to questions from the reviewers, the authors have experimented with GPT-3.5-turbo in a zero shot setting for sentiment analysis. This information can be included in the paper if it is accepted.

The reviewers ask a few other questions related to methodology/technical details, which are successfully answered by the authors. In response to one of the reviewers' observation that there are no papers from 2023 in the related works section, the authors have conducted a quick survey of work from 2023 from venues other than ACL. This information can be added in the final version of the paper.

Overall, the reviewers indicate that this is a strong and interesting paper.

---

### Decision · Program_Chairs · 2023-10-07

**Decision:**

Accept-Main

**Comment:**

The reviewers agree that this is a solid paper which contains a comprehensive evaluation. The evaluation results indicate that the proposed method works well.

The main reserve expressed by the reviewers is the limited novelty of the method. However, the authors' answer emphasise the fact that even though supervised contrastive learning has been used before, this is the first time when it is used as a representation learning tool. In response to questions from the reviewers, the authors have experimented with GPT-3.5-turbo in a zero shot setting for sentiment analysis. This information can be included in the paper if it is accepted.

The reviewers ask a few other questions related to methodology/technical details, which are successfully answered by the authors. In response to one of the reviewers' observation that there are no papers from 2023 in the related works section, the authors have conducted a quick survey of work from 2023 from venues other than ACL. This information can be added in the final version of the paper.

Overall, the reviewers indicate that this is a strong and interesting paper.